# A Polyaminobiaryl-Based β-secretase Modulator Alleviates Cognitive Impairments, Amyloid Load, Astrogliosis, and Neuroinflammation in APP^Swe^/PSEN1^ΔE9^ Mice Model of Amyloid Pathology

**DOI:** 10.3390/ijms24065285

**Published:** 2023-03-09

**Authors:** Marie Tautou, Florian Descamps, Paul-Emmanuel Larchanché, Luc Buée, Jamal El Bakali, Patricia Melnyk, Nicolas Sergeant

**Affiliations:** 1Univ. Lille, Inserm, CHU Lille, UMRS1172—LilNCog—Lille Neuroscience & Cognition, 59000 Lille, France; 2Alzheimer & Tauopathies, LabEx DISTALZ, 59045 Lille, France

**Keywords:** Alzheimer’s disease, amyloid deposits, drug development, therapeutic, neuroinflammation, astrogliosis, transgenic mice

## Abstract

The progress in Alzheimer’s disease (AD) treatment suggests a combined therapeutic approach targeting the two lesional processes of AD, which include amyloid plaques made of toxic Aβ species and neurofibrillary tangles formed of aggregates of abnormally modified Tau proteins. A pharmacophoric design, novel drug synthesis, and structure-activity relationship enabled the selection of a polyamino biaryl PEL24-199 compound. The pharmacologic activity consists of a non-competitive β-secretase (BACE1) modulatory activity in cells. Curative treatment of the Thy-Tau22 model of Tau pathology restores short-term spatial memory, decreases neurofibrillary degeneration, and alleviates astrogliosis and neuroinflammatory reactions. Modulatory effects of PEL24-199 towards APP catalytic byproducts are described in vitro, but whether PEL24-199 can alleviate the Aβ plaque load and associated inflammatory counterparts in vivo remains to be elucidated. We investigated short- and long-term spatial memory, Aβ plaque load, and inflammatory processes in APP^Swe^/PSEN1^ΔE9^ PEL24-199 treated transgenic model of amyloid pathology to achieve this objective. PEL24-199 curative treatment induced the recovery of spatial memory and decreased the amyloid plaque load in association with decreased astrogliosis and neuroinflammation. The present results underline the synthesis and selection of a promising polyaminobiaryl-based drug that modulates both Tau and, in this case, APP pathology in vivo via a neuroinflammatory-dependent process.

## 1. Introduction

Alzheimer’s disease (AD) is a neurodegenerative disease and the leading cause of dementia worldwide. The progressive extracellular secretion of toxic amyloid-β (Aβ) peptide species further framed in amyloid plaques and neurofibrillary tangles (NFTs) formed by the intraneuronal aggregation of abnormally modified Tau protein isoforms, together with neuroinflammatory processes are the major neuropathological hallmarks of AD [1,2,3]. Aβ peptides are released following successive endoproteolytic and carboxy-peptidase cleavages of their type I transmembrane Amyloid Precursor Protein (APP) [4]. The β-site APP cleaving enzyme 1 (BACE1) endoprotease cleaves APP at the first amino acid residue of the Aβ sequence, shedding the N-terminal soluble APPβ (sAPPβ) fragment and leaving the membrane-bound APP C-terminal fragment (β-CTF/C99) [5,6]. BACE2 aspartyl protease can also in Alzheimer-related conditions cleave APP at the β-site [7]. The β-CTF fragment is then processed by the γ-secretase, resulting in the release of Aβ peptide species and the APP intracellular domain (AICD), effectively settling the so-called amyloidogenic pathway [8]. Alternatively, the α-secretase can cleave APP, releasing the soluble ectodomain portion of APPα (sAPPα) and then precluding Aβ formation [9]. Aβ toxic species released in a soluble form may contribute to cognitive dysfunction at the early stages of the disease and then undergo misfolding, and oligomerization, ultimately leading to plaque formation [10].

BACE1 is a key endoprotease for generating Aβ peptides starting at position 1 or position 11 (β’-cleaving site), thus representing a rate-limiting step in the process of Aβ formation [11]. The loss of BACE1 expression in the AD mouse model (5xFAD) completely precludes amyloid deposition [12]. In contrast, in overexpressing BACE1/APPV717I double-transgenic mice, the overexpression of BACE1 significantly increased the number of diffuse and mature amyloid deposits [13]. These data, along with the studies showing that BACE expression is elevated in the frontal cortex of sporadic AD patients [14,15], suggest BACE1 as a prime target for the therapeutic perspective in AD. Over the last decades, we have seen the acute development of BACE inhibitors. Starting from peptide mimetic, more recently small chemical BACE1 inhibitors have allowed the blood-brain barrier penetrance while still achieving sufficient selectivity for BACE1 and avoiding toxic off-targets, which have proven to be challenging issues, although slight or moderate inhibition of BACE1 may be an alternative therapeutic strategy [5,16]. Actually, efforts to understand the biology of this protease have yielded evidence for multiple BACE1 targets other than APP [17]. This raises concerns about the safety of a direct BACE1 inhibition and raising the demand for an alternative approach to BACE1 indirect or non-competitive inhibition or even more selective BACE1 versus BACE2 drugs for repressing APP or redirecting cleavage by BACE1 toward the alternative β’site Glu-11 BACE1 cleavage site [18,19].

BACE1 substrates can either be processed through the secretory pathway or the early or late endolysosomal pathway, and the optimal BACE1 proteolytic activity is reached at an acidic pH [6]. Lysosomotropic drugs such as chloroquine or bafilomycin A1, a proton-pump inhibitor, repress BACE1 activity through the alkalinization of acidic cell compartments, thus leading to indirect inhibition of BACE1 proteolytic activity [20,21,22,23]. Originally derived from chloroquine [24] or amodiaquine [25], small chemical drugs have been developed. Additionally, a more recent family of polyamino biaryl compounds, including PEL24-199, derived from a bioinformatic assisted pharmacophore design, were shown to selectively repress Aβ1-x production but not the production of other Aβ_x-40/42_ species, including the Aβ_11-40/42_, suggesting a selective BACE1 inhibitory activity towards the cleavage at the first amino acid residue of Aβ sequence [26]. Moreover, in vitro β-secretase assays with our drug revealed no direct inhibitory activity of BACE1’s aspartyl protease activity, implying that our drug inhibits BACE1 indirectly or non-competitively [26]. PEL24-199 therefore selectively represses the cleavage of APP by BACE1 in the endolysosomal compartments but likely not in the secretory pathway, or even not repressing the APP cleavage at position eleven of the Aβ sequence through a yet undetermined mechanism. Therefore, our current data suggest selective repression of Aβ1-x through BACE1 inhibition at this Aβ1-x cleavage site [26].

A growing body of evidence suggests a crosstalk between the two lesions of AD pathology, Aβ peptides and Tau proteins. Indeed, it has been shown that APP metabolism regulates Tau expression, notably by inhibiting the β-secretase, which reduces the intracellular pool of Tau protein [27], and that Tau proteins deficiency rescues neuronal cell death and decreases amyloidogenic processing of APP in APP/PS1 mice [28]. High levels of Aβ induce Tau hyperphosphorylation and the formation of neurofibrillary tangles in specific brain regions in AD patients, triggering synaptic dysfunction, inflammation, and oxidative stress in affected cells recapitulated through intracerebral injection of AD-brain-derived pathological lesions [29,30,31]. Moreover, multiple studies have shown that the presence of Aβ deposition enhances trans-neuronal Tau propagation and toxicity [30,31,32]. We assume that a combined therapeutic approach targeting both Aβ and Tau may be an effective way to repress or even cure AD since both lesional processes are likely synergistic [33,34]. Thus, blocking Aβ aggregation and Tau hyperphosphorylation with small molecules could hold considerable promise as an entry point to new therapies for AD. We have previously described a novel ligand-based approach coupled with the identification of a computer-assisted pharmacophore design that allowed us to synthesize new compounds with a different scaffold [26]. Among the family of polyamino biaryl compounds, PEL24-199 (compound 31 in reference [22]) possesses a non-competitive β-secretase modulatory activity. Indeed, this compound was shown to modulate APP metabolism in vitro and more specifically to reduce Aβ_1-40_ and Aβ_1-42_ production [26]. Furthermore, we recently demonstrated that PEL24-199 could cross the blood-brain barrier and was efficient against Tau pathology in vivo [35], similar to previous drugs derived from piperazine [36]. PEL24-199 allowed the preservation of short-term spatial memory, decreased the neurofibrillary degenerating process, and reduced the astrogliosis in the Thy-Tau22 model of hippocampal Tau pathology. These beneficial effects were associated with an increase in PP2AC expression, an important phosphatase of Tau [35].

Based on the structure-activity relationship and in vivo beneficial activity of other drugs with chemical skeleton distinct from chloroquine, as well as the current pharmacological activities of PEL24-199, we hypothesize that PEL24-199’s in vivo activity against amyloid pathology may rely on chemical properties distant from the lysosomotropic or alkalizing chloroquine related properties of other drugs we evaluated. In this study, we then treated an eight-months-old APPSwe/PS1∆E9 transgenic mouse model of amyloidosis with PEL24-199 for four months in a curative paradigm to investigate whether PEL24-199 modifies the amyloid pathology in vivo. We first carried out behavioral experiments to assess if PEL24-199 caused short- and long-term spatial memory deficits. The amyloid burden and Aβ plaque load in the cortex and hippocampus of APPSwe/PS1∆E9 mice, together with APP metabolism and inflammatory responses, were investigated.

## 2. Results

### 2.1. PEL24-199 Restores the Short- and Long-Term Spatial Memory in APP/PS1 Mice

In the present experimental design, APP/PS1 mice and littermate controls were treated with PE24-199 in drinking water at 1 mg/kg until sacrifice (Figure 1A).

This curative paradigm was previously described to promote a significant benefit in a model of Tau hippocampal pathology [35,36]. Mice were treated from eight to twelve months of age after the appearance of amyloid pathology and the associated cognitive deficits reported in this transgenic model [35]. After three months of treatment, APP/PS1 mice’s motricity and anxiety were assessed with an infrared actinometer and the elevated plus maze (EPM), respectively. As shown in Figure 1B, APP/PS1 mice were more active than WT mice, but PEL24-199 had no impact on distance moved, suggesting a lack of motor modulatory activity of our drug (*p* > 0.05). Regarding basal anxiety, APP/PS1 mice were less anxious than WT mice; however, we could not find any significant impact of PEL24-199 treatment on the time spent in the open versus the closed arms (Figure 1C; *p* > 0.05), suggesting that the treatment did not affect the velocity or basal anxiety behavior in both WT and APP/PS1 mice. We then evaluated the impact of PEL24-199 on short- and long-term spatial memory using the Y-Maze and the Barnes tests, respectively. Short-term spatial memory was impaired in APP/PS1 animals, as shown by the equal time spent in the new arm versus the other arm, without distinction of preference (Figure 1D; *p* > 0.05, two-way ANOVA). The short-term memory was not modified in WT animals, as both treated and untreated WT groups showed a preference for the new arm, whereas APP/PS1 PEL24-199-treated mice spent significantly more time in the new arm than in the other arm (*p* < 0.001). The Barnes maze is a behavioral task used to measure long-term spatial memory in mice and is based on the animal’s ability to use spatial clues to find an escape hole. The parameters measured are the number of holes the rodent explores before finding the correct escape hole and the time it takes to find refuge in the target hole. Regarding the Barnes test, during the training phase, all groups exhibited a decrease in the number of total mistakes and the total latency to the target across trials (Figure 1E,F; *p* < 0.0001, two-way ANOVA). Seventy-two hours following the training phase, a probe trial was performed to evaluate the mice’s long-term spatial memory. Regardless of treatment (PEL24-199 or water), WT mice showed a clear preference for the Target quadrant versus the other zones (Figure 1G; WT untreated: *p* < 0.05; WT vs. PEL24-199: *p* < 0.0005). APP/PS1 untreated mice had no preference for the Target quadrant and exhibited long-term spatial memory deficits, as expected from this model at twelve months old (*p* > 0.05). Interestingly, the treatment with PEL24-199 significantly alleviated long-term spatial memory impairment in APP/PS1 mice, as demonstrated by a significantly increased time spent in the Target quadrant versus the other quadrants (*p* < 0.05). Altogether, these results indicate that following four months of curative treatment with PEL24-199, the long- and short-term spatial memories were restored in this APP/PS1 transgenic model of amyloidosis.

### 2.2. PEL24-199 Curative Treatment Reduces Plaque Formation and Aβ Deposition in APP/PS1 Mice Brain

In this APP/PS1 model, Aβ deposition starts at four months in the cortex and six months in the hippocampus [37]. The amyloid deposits increase in size and number with age, with a maximum reached around twelve months. To investigate Aβ protein accumulation in treated and untreated animals, we performed anti-Aβ immunohistological staining with 4G8 antibody, which detect not only mature Aβ plaques but also less condensed Aβ protein deposits (Figure 2A).

Following treatment with PEL24-199, we observed a strong reduction in the percent area covered by Aβ deposits in the hippocampus compared to the untreated APP/PS1 mice [F(1,12) = 23.02, *p* = 0.0004] (Figure 2B). Interestingly, while applying a classification of amyloid deposits according to their size, Two-way ANOVA analysis showed that PEL24-199 reduced Aβ plaques of small size, lower than 250 µm² [F(1,12) = 32.20, *p* = 0.0001], whereas larger Aβ plaques between 250–500 µm² were not affected by the treatment (*p* = 0.3601) (Figure 2C,D). To better understand the number and size of dense core mature amyloid plaques in the hippocampus and the cortex of APP/PS1 mice treated with PEL24-199, we used Thioflavin-S staining in slices from the identical individual mice used for immunostaining with 4G8 anti-Aβ antibody. As expected, the Thioflavin-S staining values were lower than the percent area covered by Aβ deposits in untreated APP/PS1 mice (Figure 2E). The number of Thioflavin-S mature amyloid plaques per mm² was highly reduced in both the cortex and the hippocampus (Figure 2F). In PEL24-199-treated animals, the percentage of the cortex and hippocampus covered by plaques was reduced roughly four-fold (Figure 2G). Overall, these data suggest that PEL24-199 treatment significantly reduces the amyloid burden and mature plaque load in the brains of APP/PS1 mice. However, we did not observe any modification of total APP, Aβ, phospho-APP at threonine 668 or APP-CTFs protein levels using the semi-quantitative western blot analysis (Figure 2H,I), implying that the PEL24-199 effect occurs through the amyloid lesional process of brain clearance of the Aβ peptide. However, Aβ plasma dosage showed no significant difference between treated and untreated APP/PS1 animals, suggesting that Aβ brain clearance was not modified by PEL24-199 (Appendix A).

### 2.3. PEL24-199 Reduces the Activation of Astrocytes and Neuroinflammation in APP/PS1 Mice

In APP/PS1 mice, increased numbers of GFAP+ astrocytes are observed after six months, especially around amyloid plaques [38]. In the curative paradigm starting at eight months, astrogliosis is already occurring in the APP/PS1 model. Since PEL24-199 was shown previously to diminish the astrogliosis in a mouse model of Tauopathy [35], we used glial fibrillary acidic protein (GFAP) immunofluorescence staining to determine the effect of PEL24-199 treatment on the astrocytic activation co-occurring with the amyloid pathology in the APP/PS1 animals (Figure 3A).

In PEL24-199-treated APP/PS1 animals, the percentage of surface in the cortex and hippocampus covered by GFAP-positive astrocyte staining was significantly reduced by more than half when compared to untreated animals (Figure 3B). Likewise, immunohistological staining revealed a significant decrease in the percent area covered by activated astrocytes in the hippocampus of APP/PS1 mice treated with PEL24-199 (Figure 3C,D). The GFAP staining alone is not sufficient to ascertain the reactivity of astrocytes. Therefore, we used qPCR to investigate the additional astrocytic markers reported to be increased in reactive astrocytes [39], together with several microglial activation markers. We showed that most of the markers were significantly decreased in the cortex of PEL24-199 treated animals compared to untreated APP/PS1 mice. However, the most significant changes were observed in the hippocampus, in which ten out of the eleven markers tested were decreased, among which five were significantly reduced (Figure 3E). GFAP mRNA was also significantly decreased, as were two other reactive astrocytes secreted protein transcripts, C3 and Lcn2 (Figure 3F–H). Moreover, two microglia markers of neuroinflammation, TLR2, and Clec7a [40,41] were also significantly reduced in animals treated with PEL24-199 (Figure 3I,J). Following the previous results obtained, GFAP protein levels were also decreased both in the hippocampus and in the cortex of APP/PS1 mice treated with PEL24-199 (Figure 3K,L). Overall, our results suggest that astrogliosis and microglial gene activation in APP/PS1 mice are significantly reduced by curative PEL24-199 treatment, and both processes might contribute to the rescue of the amyloid pathology and memory impairment in our PEL24-199-treated APP/PS1 mice.

### 2.4. PEL24-199 Treatment Increases Phosphatase PP2A and GGA1 Protein Levels in APP/PS1 Mice

As the principal Tau serine/threonine phosphatase PP2A [42] was increased in a model of hippocampal tauopathy under PEL24-199 treatment [35], we investigated by western blot if this phosphatase protein level was similarly increased in the APP/PS1-treated animals (Figure 4A). 

Indeed, a four-month treatment sharply increased the expression of the catalytic subunit PP2AC in the hippocampus and the cortex of the animals (Figure 4B), following the obtained results of the tauopathy model. We then investigated the protein level of kinases Cdk5 and GSK3β, involved in Tau [43,44,45] and APP hyperphosphorylation [46,47] and associated pathologic changes. Cdk5 and GSK3β protein levels remain unchanged both in the hippocampus and the cortex of APP/PS1 mice treated with PEL24-199 (Figure 4C,D). This suggests that kinase expression is not modified by the PEL24-199 treatment. The expression of Fyn protein kinase, which has been linked to Aβ synaptic toxicity and Tau phosphorylation [45,48], was also investigated. The protein levels of Fyn were unchanged under PEL treatment conditions (Figure 4E). 

BACE1 expression, cell localization, and activity are controlled by several factors. Since PEL24-199 has a non-competitive β-secretase activity in a cell system, we investigated whether our drug treatment did not affect the synthesis of GGA1 protein, a key β-secretase regulating factor. Golgi-localized, gamma-ear-containing, ADP ribosylation factor-binding (GGA) proteins have been shown to interact with the cytoplasmic domain of BACE1 in a phosphorylation-state-dependent manner and to regulate the endocytic trafficking of this protease [49]. This alters the proteolytic processing of APP [50]. GGA1 is thus essential to mediate the rapid trafficking of phosphorylated BACE1 to recycling endosomes and limit the production of toxic metabolites [51]. In this experiment, PEL24-199 treatment significantly increased GGA1 protein expression levels in the hippocampus but not in the cortex (Figure 4F), indicating a potential modulation of trafficking or regulation of BACE1 in vivo. However, more research will be needed to determine the mechanism of GGA1 upregulation and BACE1 trafficking and activity.

## 3. Discussion

In the present study, we show that the β-secretase non-competitive inhibitor compound PEL24-199 reduces amyloid pathology in the APP/PS1 model via a process that includes astrogliosis and neuroinflammation. Our results show that a curative treatment of four months in eight-month-old APP/PS1 mice with PEL24-199 restores short- and long-term spatial memory and decreases amyloid pathology and neuroinflammation in this transgenic lesional model of amyloid deposition. This curative effect is also accompanied by an increase in PP2A catalytic subunit and an upregulation in GGA1 protein levels, as previously shown.

The results of the APP/PS1 model show a highly significant increase in PP2Ac phosphatase. This confirms the data from the THY-Tau22 model and shows that PP2A appears to play a major role in the beneficial curative effects mediated by PEL24-199 [35]. This supports other studies highlighting that increasing PP2A activity protected against Aβ peptide-induced dysfunction [52] and decreased overproduction of Aβ fragments [53]. Furthermore, repressing PP2A silencing inhibitor-2 resulted in long-term attenuation of amyloidogenesis in Tg2576 mice by inhibiting APP hyperphosphorylation and β-secretase activity [54]. Increasing PP2A activity also promotes non-amyloidogenic APP processing in a Fyn-dependent manner [55]. In this study, the level of Fyn protein synthesis was not altered but a modified cellular localization or phosphorylation or activity cannot be ruled out and would need further investigation. An increase in PP2A thus appears to selectively affect the pathological response to elevated Aβ levels, allowing the cellular system to mitigate the toxicity of the amyloid pathology. Our results showed that PEL24-199 also increased the activity of another BACE1 regulatory-associated protein, GGA1, in the hippocampus of treated APP/PS1 mice. GGA1 is involved in addressing β-secretase in endosomes and allowing rapid transduction of β-secretase from early endosomes; the cell vesicle compartment where β-secretase processing of APP processing also occurs and is also a rate-limiting compartment for Aβ peptide generation. Thus, overexpression of GGA1 reduces APP cleavage by the β-secretase and, consequently, the secretion of Aβ peptides. Moreover, the modulation of APP processing by GGA1 is independent of the direct interaction between GGA1 and BACE1 [49,50,56]. Thus, GGA1 proteins are suggested to be involved in the pathophysiology of AD concerning amyloid pathology, for which GGA1 expression was shown to be decreased in human AD brains. GGA1 could contribute to the beneficial effect observed after PEL24-199 treatment via the molecule’s action on GGA1 expression or degradation, allowing spatial segregation of β-secretase and its substrate APP. This modifies the routing of BACE1 and decreases the production of Aβ peptides. Neither the BACE1 enzymatic assay nor BACE1 brain tissue expression provides clues to decipher the mechanism of PEL24-199 β-secretase modulatory activity determined in SY5Y-APP cells [26]. Further investigation would be needed to determine the mode of action of PEL24-199 toward the modulatory activity of BACE1. This modulatory activity could occur toward a preferential substrate recognition, β-site versus β’-site, through its cell-trafficking implicating GGA1, post-translational modifications (phosphorylation in relationship with PP2A expression), or co-factors, which remain currently ill-defined.

Surprisingly, treatment with PEL24-199 decreased the number of small amyloid plaques (between 50 µm² and 250 µm²), with no apparent change in larger plaques (250 µm²–500 µm²). Thioflavin-S labeling revealed that large and mature amyloid deposits were reduced after the PEL24-199 treatment, while APP metabolites expression remained unchanged. PEL24-199 also did not influence total APP levels or its phosphorylation on threonine 668 [46,47]. Together, these observations suggest that PEL24-199 acts on developing lesions by decreasing the number of small amyloid plaques and on existing mature Aβ plaques by promoting the clearance of amyloid plaques, thus allowing an overall decrease in the number of amyloid deposits. The overall amount of Aβ in contrast remains stable. However, decreased amyloid plaque may not be systematically associated with decreased Aβ peptide levels. The global amount of Aβ peptides is the sum of the Aβ produced and the Aβ peptides released from the clearance of Aβ deposits. An equilibrium between these two processes could be illustrated by the maintenance of Aβ peptide levels since a difference cannot be made. Aβ brain clearance occurs in part through the lymphatic and brain blood barrier systems, leading to modulation of plasma Aβ concentration. To address this possibility, we measured the concentration of Aβ peptides in plasma. Quantification of Aβ_1-40_ and Aβ_1-42_ levels in the blood of the animals showed that PEL24-199 decreased in a non-significant extent Aβ peptides plasmatic levels in PEL24-199-treated APP/PS1 mice (Appendix A), which then potentially ruled out a higher clearance of the brain Aβ through either the lymphatic to blood or brain blood barrier-dependent mechanisms.

Alternatively, we hypothesized that PEL24-199 could promote the removal of amyloid plaques and abnormal Tau proteins by promoting their phagocytosis by astrocytes and microglia [39,40,41,57]. Reactive astrocytes with increased GFAP expression have been shown to surround amyloid plaques in the brains of AD patients [38]. As early as six months, APP/PS1 mice have increased GFAP expression, density, the area occupied by GFAP+ astrocytes, the expression of more reactive astrocyte-related genes, and hypertrophied astrocytes surrounding Aβ plaques [37]. Interestingly, our results showed that PEL24-199 caused GFAP gene expression, GFAP protein expression, the area occupied by GFAP+ astrocytes, and so-called reactive astrocytes to decrease significantly in both the cortex and hippocampus of APP/PS1 mice. To confirm these results, we investigated the effect of PEL24-199 treatment in APP/PS1 mice on the expression of several specific markers of reactive astrocytes, such as vimentin, connexin43, C3, and Lcn2. The expression of all these markers was decreased in the hippocampus of PEL24-199-treated mice, among which C3 and Lcn2 were significantly reduced. C3 protein is found highly increased in the brains of human AD patients and in different APP transgenic mouse models [39]. C3 is an astrocytic target of NFκB, activated by exposure to Aβ peptides, which will cause altered dendritic morphology and neuronal function. This protein is thought to contribute to Tau pathology-induced neurodegeneration and plays an important role in the glia response to Aβ plaques [58,59]. C3 inhibition has been confirmed by others to reduce the number of pro-inflammatory microglia, decreasing synaptic and neuronal loss [60,61]. Lcn2 is a protein whose expression also increases in AD lesion-affected brain regions in patients and has recently been identified as a risk factor contributing to the development and progression of AD [62]. It appears that Lcn2 sensitizes neuronal cells to Aβ peptide toxicity, in addition to impeding the neuroprotective action of TNF-α in the early stages of the disease [63]. Gene expression of these two astrocytic markers that are central to AD lesions was thus reduced by curative treatment with the molecule PEL24-199. Another important factor of neuroinflammation is microglia. We then investigated the potential effects of PEL24-199 on these immune cells through several gene transcripts characteristic of reactive microglia, such as C1qa, Clec7a, Itgax, and TLR2. Our data show that the expression of these genes was reduced in the hippocampus of mice treated with PEL24-199, except for C1qa. The highest modulatory effect of PEL24-199 was observed for the Clec7a and TLR2 genes, which significantly decreased following treatment. TLR2 is a microglial receptor important for the initiation of neuroinflammation by Aβ peptides and is found overexpressed in AD [40]. In APP/PS1 models, TLR2 inhibition reduces glial cell reactivity, reduces Aβ deposition, and improves cognitive function [64]. These results are supported by those of other teams showing that TLR2 receptor deficiency decreases Aβ_1-42_-induced oxidative damage and inflammation and increases Aβ peptide clearance [65,66]. Furthermore, TLR2 appears to be paramount in determining the neuroprotective or neurotoxic profile of microglia [67]. Clec7a is disease-associated microglia (DAM) gene overexpressed in APP/APOE and APP/PS1 models [68,69]. Clec7a has been identified as a key player in the transition to a reactive microglia phenotype, associated with neurodegeneration and aging, and an antibody against Clec7a rescues microglia activation [70]. The reduction in Aβ peptide deposition observed with PEL24-199 treatment could thus be a secondary consequence of the reduction in astrogliosis and microgliosis, as both astrocytes and activated proinflammatory microglia have been shown to exacerbate the number of amyloid plaques and thus maintain the amyloid and Tau pathologies. Several studies demonstrate that plasma GFAP concentrations are elevated in cognitively normal older adults at risk for AD [71] thus making plasma GFAP a potential theragnostic marker. These observations then support the hypothesis that astrocytic activation begins in the pre-symptomatic stage of AD and is associated with brain Aβ load [72]. Early stage Aβ deposits could thus interact with the inflammatory response, indicating astrocytosis as a driving force with an early contribution to Alzheimer’s physiopathology [72]. Recently, it has been shown that GFAP could be an early marker of AD and possesses a strong correlation with Aβ peptide concentration in cerebrospinal fluid [73]. Because neurotoxic reactive astrocytes are induced by the activation of microglia, microglia might even be an earlier marker [74]. Helping to keep or bring glial cells back into this clearance and phagocytosis stage before the activation and pathology worsening phase, as the PEL24-199 molecule seems to do, could thus be an interesting therapeutic strategy. These results are consistent with a recently published study showing that AZP2006, a related compound, possessed neuroprotective properties in two animal models of AD and aging [75]. Thus, the beneficial effects of AZP2006 were linked to the inflammation regulator progranulin and the inhibition of TLR9 receptors that are normally responsible for proinflammation when activated.

## 4. Materials and Methods

### 4.1. Animals

We used the well-characterized heterozygous male APP^Swe^/PSEN1^∆E9^ (herein referred to as APP/PS1, C57Bl6/J background) and littermate controls [36]. All animals were housed in a pathogen-free facility on a 12/12-h light-dark cycle and maintained at a constant temperature of 22 °C with ad libitum access to food and water. Animals were maintained at five to six per cage (Tecniplast Cages 1284L, Tecniplast, Amiens, France) with genotype segregated, and enrichment in the form of a small cylinder ‘‘cocoon’ and a transparent plastic hut (Techniplast SAFE hut). The animals were used in compliance with European standards for the care and use of laboratory animals, and the experiments conducted in this study were authorized by the French Direction of Veterinary Services with the approved registration number APAFIS#10392-201706231206250v4.

### 4.2. Drug Treatment

PEL24-199 compound was synthesized according to the previously described procedure [26]. For the present study, animals (minimum of nine animals per condition) were randomly distributed, and APP/PS1 and WT mice were treated for 4 months, starting at 8 months of age. PEL24-199 treatment was delivered in the drinking water at a final concentration of 1 mg/kg, i.e., 12.5 μg/mL into the drinking water considering an average weight of 25 g/mouse drinking 4 mL of water per day. Opaque black drinking bottles (Tecniplast, Amiens, France) were changed once per week even if aqueous solutions of compound PEL24-199 were previously demonstrated to be stable in the dark for more than one week. A one-month safety pilot study in WT animals was conducted to determine the innocuousness of compounds PEL24-199 at doses of 1 mg/kg and 5 mg/kg and to determine whether PEL24-199 could cross the blood-brain barrier [36]. The volume of solution consumed by the mice was measured throughout the treatment period. The animal treatment was performed until the sacrifice of the animals.

### 4.3. Behavioral Tests

#### 4.3.1. Actimetry

A minimum of nine mice per group were used for every behavioral test. Mice were placed in the center of an infrared Actinometer (45 × 45 × 35 cm; Bioseb), composed of a two-dimensional square frame, and left free to explore the arena for 10 min. The spontaneous behavior of mice was tracked, and the distance moved, and the velocity was recorded by Actitrack software (BioSeb, Vitrolles, France).

#### 4.3.2. Elevated Plus Maze

Basal anxiety, which could interfere with spatial memory assessment, was evaluated in treated and untreated animals using the elevated plus maze (EPM) test. Mice were placed in the center of a plus-shaped maze consisting of two 10-cm-wide open arms and two 10-cm-wide enclosed arms elevated 50 cm above the floor surface. Parameters including the number of entries into each arm, time spent in the open versus the closed arms, and percentage of open arm entries were measured for 5 min and acquired by video recording using EthoVision video tracking equipment and software (Noldus Information Technology, Paris, France) in an animal facility dedicated room.

#### 4.3.3. Y-Maze

Short-term spatial memory was assessed using the Y-maze test. The Y-maze consists of three 10-cm-wide enclosed arms surrounded by four different spatial clues. One of the two arms opposite the starting (S) arm was alternately closed during the learning phase. Each mouse was positioned in the starting arm and was free to explore the maze for 5 min. Then, during the retention phase of 2 min, the mouse was returned to its home cage. During the memory test phase of 5 min, the closed arm was opened, and the mouse was placed in the starting arm. The previously closed arm was named the « New arm » and the already opened arm was named « Other arms ». Parameters—total distance traveled, velocity, the alternation between the arms, entries into the three arms—were video acquired during 5 min. The short-term spatial memory test was considered successful when the proportion of entries and time spent in the new arm was higher than the time spent in the other arms.

#### 4.3.4. Barnes Maze

Long-term spatial memory was assessed using the Barnes maze test [76]. The Barnes maze is a white circular PVC open platform surface (120 cm in diameter) with forty equally spaced holes (5 cm in diameter) located 5 cm from the edge and an opaque black escape box located under one of the 40 holes. The platform is placed in the center of the room, elevated 80 cm above the floor surface, enlightened by an intense ceiling light (800 lux), and surrounded by four different distant spatial visual clues. The day before the training, mice completed a habituation trial to become familiar with the maze environment and practice entering the escape box. Mice are allowed to freely explore the platform and the escape box for 5 min. Then, mice completed four days of acquisition training with four trials per day and three inter-trial intervals of 15 min. At the start of each trial, the mice were placed in a gray PVC start chamber located in the center of the maze. After 15 s, the start chamber is lifted, and the mouse is trained to locate the escape hole (randomized for all mice) using spatial visual clues surrounding the maze. If a mouse did not enter the escape hole within 3 min it was gently pulled to the escape hole and remained 1 min in the escape box before being returned to the home cage. For each trial, the number of total mistakes and total latency to enter the escape box were recorded using the Ethovision XT tracking system (Noldus). To reduce intra-maze odor cues, the maze was rotated clockwise a quarter turn every day, and the maze surface and escape box were cleaned with ethanol following each trial. Seventy-two hours after the last day of training, mice completed a 2 min probe trial where the escape box was removed. The time spent in the Target quadrant (“Target quadrant”) versus the time spent on average in non-target quadrants (“Others”) was determined following video acquisition using the Ethovision XT tracking system (Noldus).

### 4.4. Sacrifice and Brain Tissue Preparation

The mice were sacrificed by beheading, and the blood was collected from the neck in heparinized tubes. Brains were removed, and one hemisphere was post-fixed for immunohistochemistry in 4% paraformaldehyde fixative diluted in PBS (pH 7.4) for a week at 4 °C and transferred to a 20% sucrose solution overnight before being frozen. The cortex and hippocampus of the other half of the brain were dissected, disposed of in a 1.5 mL isopropylene tube, and then snape-frozen by immersion of the tube in an isopropanol solution added with dry ice. Brain tissue was stored at −80 °C until used for biochemical analyses. Cortex and hippocampus were added to 200 µL of Tris-Sucrose buffer (TSB) [Tris-HCl 10 mM, pH 7.4 with 10% sucrose (*w*/*v*)] and tissue was sonicated (40 pulses, amplitude 60, 0.5 kHz). A BCA assay was used to determine the total protein concentration in the TSB lysate of the hippocampus and the cortex, and each brain lysate protein concentration was diluted to a final concentration of 1mg/mL with the addition of TSB.

### 4.5. SDS-PAGE and Western Blot

The TSB brain lysate samples were added to one volume of NuPAGE LDS 2X sample buffer supplemented with 20% NuPAGE reducing agents (Invitrogen, Villebon-sur-Yvette, France) and heated for 10 min at 70 °C. Using the precast 4–12% Criterion™ XT Bis-Tris polyacrylamide eighteen-well gels (Bio-Rad, Marnes-la-Coquette, France), 15 µg of total proteins were loaded per well, and protein electrophoresis was achieved by applying a tension of 200 V during 60 min using the CriterionTM Electrophoresis Cell using the NuPAGE MOPS SDS running buffer (1X). Apparent molecular weight calibration was achieved using molecular weight markers (Novex and Magic Marks, Life Technologies, Villebon-sur-Yvette, France). Following electrophoresis, proteins were transferred from the polyacrylamide gel onto a nitrocellulose membrane of 0.4 μM pore size (G&E Healthcare, Buc, France) using the Criterion™ blotting system by applying a constant tension of 100 V for 40 min. For APP metabolites detection (Aβ and CTFs), 16.5% CriterionTM XT Bis-Tris polyacrylamide eighteen-well gels (Bio-Rad) were used with the Tris-Tricine running Buffer (BioRad) and electrophoresis was achieved by applying a tension of 150 V for 2 h. Proteins were transferred onto a nitrocellulose membrane of 0.2 µm pore size (G&E Healthcare) using the Criterion™ blotting system by applying a tension of 100 V for 30 min.

Protein transfer and quality were determined by a reversible Ponceau Red coloration (0.2% xylidine Ponceau Red and 3% trichloroacetic acid) before an extensive wash under deionized water. Membranes were blocked for 1 h in 25 mM Tris–HCl pH 8.0, 150 mM NaCl, 0.1% Tween-20 (*v*/*v*) (TBS-T), and 5% (*w*/*v*) of skimmed milk (TBS-M) or 5% (*w*/*v*) of bovine serum albumin (TBS-BSA) depending on the antibody. Membranes were then incubated with primary antibodies overnight at 4 °C. Conditions of use of the primary and secondary antibodies are summarized in Appendix A. Membranes were rinsed three times for 10 min with TBS-T and then incubated with secondary antibodies for 45 min at room temperature (RT). The immunoreactive complexes were revealed using either the ECL™ or ECL™ Prime (Cytiva, Velizy-Villacoublay, France) following the manufacturer’s instructions, and western-blot images and signals were acquired with the LAS-3000 system (FujiFilm, Busy-Saint-Georges, France). Quantifications of protein expression were calculated with ImageQuant™ TL Software (version 10.2, Cytiva) and values for each sample were divided by the quantified values of GAPDH staining. The semi-quantitative values for immunodetected samples of the treated conditions were divided by the semi-quantitative values of the control samples to express the results as the percentage of the untreated condition. An average of a minimum of six individual mouse brain protein lysates were used for these experiments, and one antibody staining was performed by gel in addition to the GAPDH secondary labeling.

### 4.6. Immunohistochemistry

Antibodies used in this study are listed in Appendix A. Coronal free-floating brain sections of 40 µm were obtained with a cryostat (CM3050 S, Leica Wetzlar, Germany). The sections of the hippocampus were selected according to the stereological rules and stored in PBS (phosphate buffer saline) with 0.2% sodium azide in twelve-well plastic plates at 4 °C. For Aβ immunohistochemistry, sections were washed with 0.2% Triton X-100 PBS buffer for permeabilization. Sections for Aβ experiments were incubated with a 0.3% hydrogen peroxide solution (1:100; #16911, Sigma, Saint-Quentin-Fallavier, France) and pretreated with an 80% formic acid (*v*/*v*) solution for 3 min. Sections were then blocked with normal goat serum (1:100; S1000, Vector Laboratories, Newark, CA 94560, USA) diluted in PBS for 1hr before incubation with a mouse biotinylated anti-Aβ antibody (4G8) at 4 °C overnight. For GFAP immunohistochemistry, sections were washed with 0.2% Triton X-100 PBS buffer for permeabilization, then incubated with a 0.3% hydrogen peroxide solution (1:100; #16911, Sigma), and then blocked with 5% normal goat serum (S1000, Vector Laboratories) diluted in PBS for 1hr. Sections were then incubated with rabbit anti-GFAP antibodies at 4 °C overnight before being incubated with the secondary anti-rabbit biotinylated antibody (BA-1000, Vector Laboratories) for 1 h. After washing in 0.2% Triton X-100 in PBS, all the sections were incubated with the ABC kit (Vector Laboratories) for 2 h and developed using DAB (Sigma) before being rinsed with physiological serum. Brain sections were mounted on glass slides (Superfrost Plus, ThermoScientific, 4817 BL Breda, The Netherlands) and dehydrated in sequential baths of 30%, 70%, 95%, and 100% (*v*/*v*) ethanol for 5 min each. Then the slides were immersed in toluene for 15 min and fixed with mounting medium (VectaMount Permanent Mounting Medium H-5000, Vector Laboratories) and glass coverslips. Images were acquired using Zeiss Axioscan.Z1 slidescan and quantification of the number of plaques, discrimination by plaques size, and area covered by GFAP staining in the hippocampus of APP/PS1 mice were performed using Image J and a custom macro. The custom macro derived from the Image J Immunohistochemistry Image Analysis Toolbox was used (https://imagej.nih.gov/ij/plugins/ihc-toolbox/index.html, accessed on 12 January 2021). After adopting the DAPI contrast, the region of interest was manually defined. Blind to the images’ identification, tests were run on a few representative Aβ-labeled and Thioflavine-stained images to define the threshold to apply that encompasses the greatest number of plaques without including non-specific signals. The same procedure was run for the GFAP astrocyte staining. These defined thresholds were applied to all images and executed automatically. The ten measured parameters were as follows: Total surface of the area in mm², Total number of plaques, Plaques total surface in mm², Number of plaques 2–150 µm², Number of plaques 150–500 µm², Number of plaques 500–1500 µm², Ratio plaques area per mm², Ratio number of plaques per mm², GFAP total surface in mm², Ratio GFAP area per mm².

For immunofluorescence studies, coronal brain sections were permeabilized with 0.2% Triton X-100 in PBS and blocked with normal goat serum (1:100; S1000 Vector Laboratories) diluted in PBS for 1hr before incubation with an anti-GFAP antibody at 4 °C overnight. After washes, sections were incubated with secondary antibody AlexaFluor 568 goat anti-rabbit IgG (A11011; Thermo-Fisher) in 0.2% Triton X-100 in PBS for 1 h. Sections were incubated with DAPI (1:5000; Sigma-Aldrich) for 5 min and mounted on glass slides. To visualize mature amyloid plaques, sections were incubated with a 1% Thioflavine-S in water solution for 30 min and then washed with sequential 80% and 90% ethanol baths. Sections were finally treated with 0.3% Suden Black (Merck Millipore 2160) for 5 min and washed with 70% ethanol to block autofluorescence. Images were acquired using a Fluorescence Zeiss Axioscan.Z1 slidescan (Carl Zeiss, Rueil-Malmaison, France). Quantification of GFAP staining and amyloid plaques was performed using Image J and an IHC ImageJ custom macro. The number of plaques and the area covered by amyloid plaques were calculated in the cortex and hippocampus of the APP/PS1 mice. An average of six anteroposterior sections were selected according to the Allen mouse brain atlas in four mice on average and for each experimental condition.

### 4.7. mRNA Extraction and Quantitative Real-Time PCR Analysis

Total RNA was extracted from hippocampi and cortex mouse brain tissue and purified using the RNeasy Lipid Tissue Mini Kit (Qiagen, Les Ulis, France) following the manufacturer’s instructions. Extracted RNAs were quantified by the NanoDropOne spectrophotometer (Thermo Fisher Scientific) and the absorbance ratios at 260/280 nm and 260/230 nm were measured to determine the RNA purity. A total of 500 ng of total RNA were reverse-transcribed using the High-Capacity cDNA reverse transcription kit (Applied Biosystems, 4368814, Villebon-sur-Yvette, France) in a final reaction volume of 20 µL. The following conditions were applied to the thermocycler: 10 min at 25 °C, 120 min at 37 °C followed by 5 min at 85 °C. The RT–qPCR reactions were performed in a Real-Time PCR System StepOnePlus (Applied Biosystems). Each reaction was performed in duplicate in 96-well plates (Applied Biosystems) with a final reaction volume of 10 μL. All reactions contained 2 μL of cDNA and 8 µL of SybrGreen mix (5 μL of Power SYBR Green PCR Master Mix (Applied Biosystems), 0.1 μL of each primer, and 2.8 μL of DEPC-treated water). The reaction protocol starts with a 10 min initial denaturation step at 95 °C, followed by forty cycles of 95 °C for 15 s, then a step at 60 °C for 25 s. Subsequently, the melting curve was verified by amplification of a single product, which was generated starting at 95 °C for 15 s, then 60 °C for 1 min and increasing by 1 °C every minute to reach 95 °C during 15 s. Each experiment included negative template controls and internal controls. Cyclophilin A was used as a reference gene, and the relative expression of target transcripts was determined by the ∆∆CT method. The sequences of primers used are given in Appendix A. A minimum of ten mice per group were used for these experiments.

### 4.8. ELISA

#### Aβ_1-40_ and Aβ_1-42_ Levels in Plasma

Blood samples in heparinized tubes were centrifuged at 10,000 rpm for 15 min (Centrifuge 5424R, Eppendorf, Montesson, France), and plasma was collected. Plasma levels of Aβ_1-40_ and Aβ_1-42_ were measured using ELISA kits (Plasma Beta-Amyloid 1-40 ELISA, EuroImmun, EQ6511-9601, and Plasma Beta-Amyloid 1-42 ELISA, EuroImmun, EQ6521-9601, EuroImmun, Bussy-Saint-Martin, France) following the manufacturer’s instructions. Briefly, 25 µL of the samples were diluted in 175 µL of the dilution buffer. Then, 20 µL of biotin solution per well was incubated with 80 µL of the diluted samples, calibrators, and controls for 3 h at room temperature. The ELISA plate was washed using the washing buffer, and 100 µL per well of enzyme conjugate was added for 30 min. The wells were washed again, 100 µL per well of chromogen/substrate was added and the plate was incubated for 30 min in the dark. 100 µL of stop solution per well was added to stop the reaction, and the absorbance at 450 nm was measured with a Multiskan Ascent spectrophotometer (ThermoLab Systems, Issy-les-Moulineaux, France). The amounts of Aβ_1-40_ and Aβ_1-42_ in the plasma were estimated by reference to the manufacturer’s standard curve and expressed in pg/mL of plasma. A minimum of seven mice per group were used for this experiment.

### 4.9. Statistics

Image acquisitions and quantifications, as well as behavioral evaluations, were performed by investigators blind to the experimental condition. Results are expressed as the means ± SEM. Differences between mean values were determined using the Student’s *t*-test or One-Way ANOVA using Graphpad Prism Software v8 (Boston, MA 02110, USA). Multiple comparisons between data series were made using Two-Way ANOVA. *p* values < 0.05 were considered significant.

## 5. Conclusions

Our results show that a curative treatment of APP/PS1 mice with PEL24-199 restores short- and long-term memory and alleviates the amyloid pathology in this mouse model, as summarized in Figure 5. We show that this rescuing effect is accompanied by reduced astrogliosis and neuroinflammation and an increase in PP2AC expression levels. However, limitations of the present study relate to limited mechanistic information providing a potential relationship between PEL24-199 cell activity and the in vivo effect against both lesional processes of AD. Although alkalizing or lysosomotropic properties appear dispensable, the mechanism related to BACE1 modulatory inhibition would need further investigation to exclude a yet undetermined and unrelated mechanism targeted by PEL24-199. Overall, these results demonstrate the promising potential of small molecules in an astrocytes-mediated pharmacological intervention strategy that could improve AD deficits even after the onset of the symptomatic phase of the disease.

## Figures and Tables

**Figure 1 ijms-24-05285-f001:**
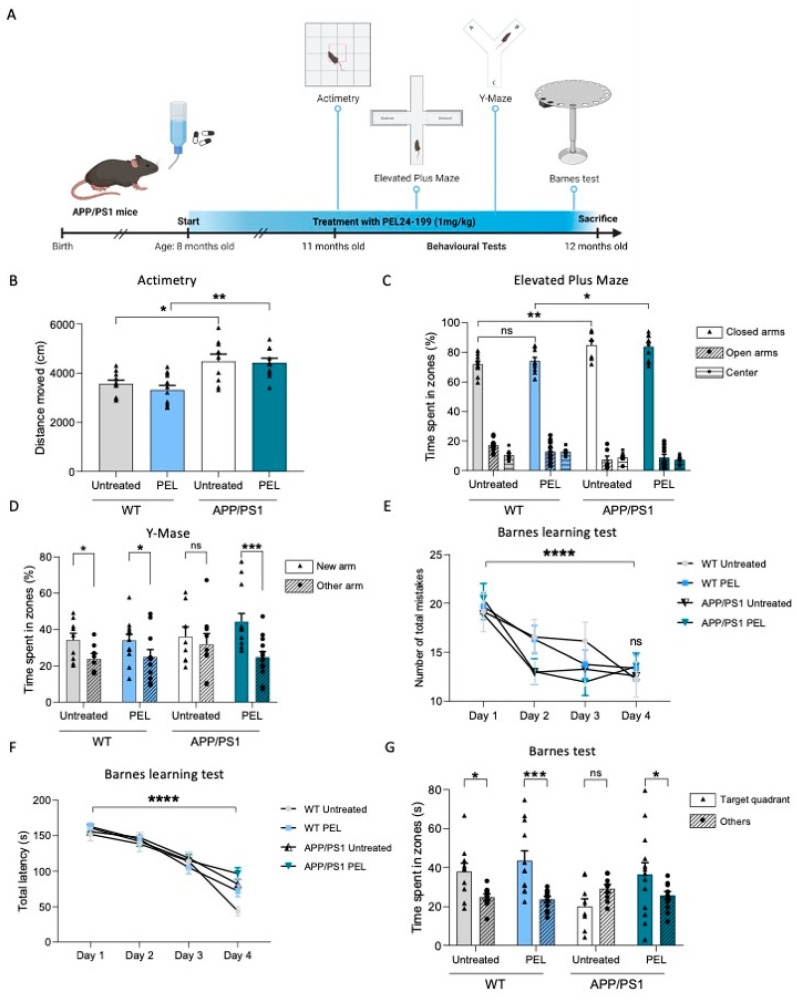
PEL24-199 restores short- and long-term spatial memory in APP/PS1 mice. (**A**) Time representation of APP/PS1 mice treatment and behavioral testing. The mice were treated for four months starting at eight months old, with 1 mg/kg of PEL24-199. Behavioral tests (Actimetry, Elevated Plus Maze, Y-Maze, and Barnes test) were performed at eleven months old, and the mice were sacrificed at twelve months old. (**B**) Locomotion and motricity tests of untreated versus PEL24-199-treated WT and APP/PS1 animals. Histograms represent the means ± SEM of the distance moved during 10 min. (**C**) Anxiety test of untreated versus PEL24-199-treated WT and APP/PS1 animals. Histograms represent the means ± SEM of the time spent in the open arms (triangle) versus the closed arms (dots and hatched bars). (**D**) Short-term memory test of untreated versus PEL24-199-treated WT and APP/PS1 animals. Histograms represent the means ± SEM of the time spent in the new arm (triangle) versus the other arm (dots and hatched bars). (**E**,**F**). Learning phase of the long-term memory test between Day one and Day four of untreated versus PEL24-199-treated WT and APP/PS1 animals, as indicated by the total latency and the number of total mistakes needed to find the target hole. (**G**). Long-term spatial memory was assessed 72 h after the last day of the learning phase. Results represent the percentage of time spent in the target quadrant versus non-target quadrants (others). WT mice (both treated with water or PEL24-199) spent significantly more time in the target quadrant, indicative of a preserved spatial memory. While APP/PS1 mice exhibited spatial memory deficits, as underlined by their lack of preference for the target quadrant, PEL24-199-treated APP/PS1 mice behaved as WT mice, suggesting that the treatment rescued the memory impairment. * *p* < 0.05, ** *p* < 0.01, *** *p* < 0.001, **** *p* < 0.0001, and *ns* as not significant. using Two-way ANOVA; n = 8–12 per group.

**Figure 2 ijms-24-05285-f002:**
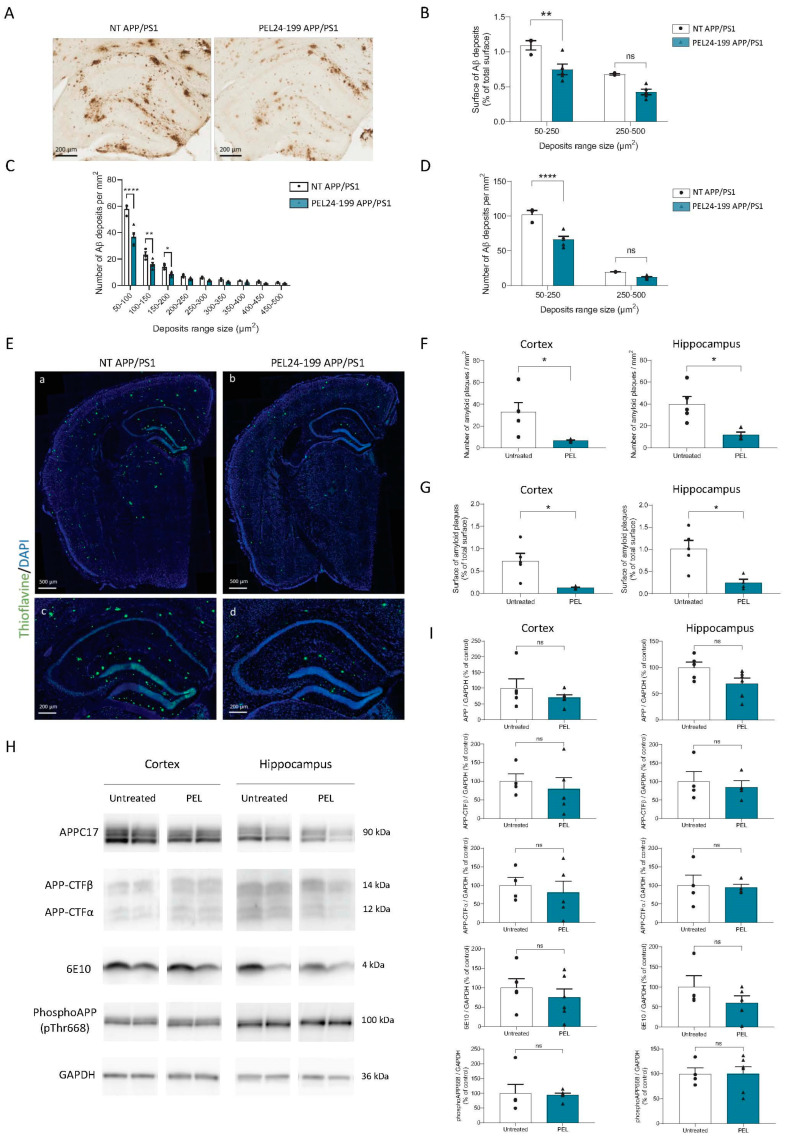
PEL24-199 decreases Aβ deposits and amyloid plaque burden in APP/PS1 mice. (**A**) Representative images of 4G8 staining in the hippocampus of a twelve months-old APP/PS1 treated with water or PEL24-199. Scale bar = 200 µm. Aβ deposit distribution (**B**), size, and number (**C**,**D**) were examined in the hippocampus and the cortex of APP/PS1 mice treated or not with PEL24-199. We found that the treatment with PEL24-199 significantly reduced both in the cortex and hippocampus the surface of Aβ deposits (**B**) and the density of plaques of lower size (between 50–250 µm^2^) as compared with APP/PS1 untreated animals (50–250 µm^2^, *p* < 0.0001 using Two-Way ANOVA; n = 4–5 per group) (**C**–**E**) Representative images of Thioflavin-S (green) and DAPI (blue) staining in the hippocampus of a twelve months-old APP/PS1 treated with water or PEL24-199. Upper panels: scale bar = 500 µm; Lower panels: scale bar = 200 µm. The number (**F**) and surface (**G**) of Thioflavin-S-positive plaques per mm² were investigated in the cortex and hippocampus of APP/PS1 mice treated or not with PEL24-199. Both in the cortex and hippocampus, the surface and a number of amyloid plaques were significantly decreased. Significance at the Mann-Whitney statistical test is indicated by *: *p* < 0.05, ** *p* < 0.001, **** *p* < 0.0001, and *ns* as not significant. (**H**,**I**) Western blot analysis was performed in the cortex and hippocampus of water, and PEL24-199-treated APP/PS1 mice. The analysis did not reveal any change in total APP, Carboxy-terminal fragments (CTFs), the Aβ peptide, or phosphorylated APP at Thr668 protein levels (n = 5–6 per group). ns indicates no significance in the Mann-Whitney statistical test.

**Figure 3 ijms-24-05285-f003:**
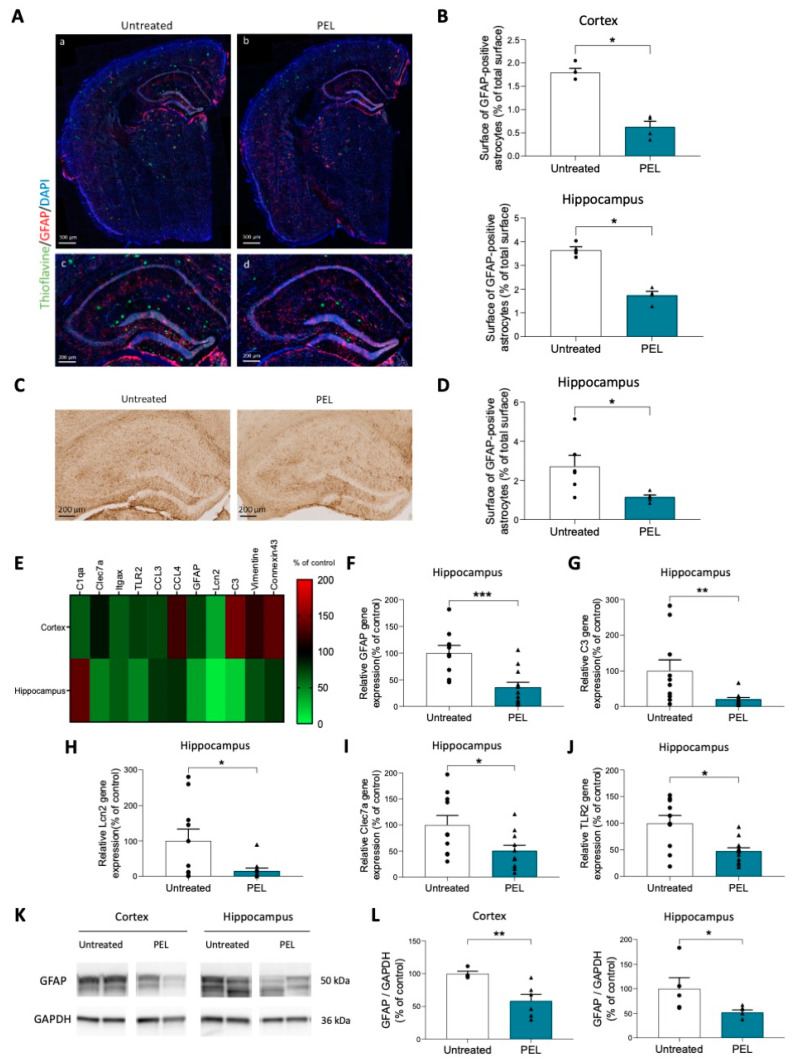
Astrogliosis and inflammation are decreased by PEL24-199 treatment. (**A**) Representative images of the cortex and hippocampus staining for the Thioflavin-S (green), DAPI (blue), and the reactive astrocyte marker GFAP (red). Upper panels: scale bar = 500 µm; Lower panels: scale bar = 200 µm. (**B**) The percentage of a total surface of GFAP-positive astrocytes was investigated in the cortex and hippocampus of APP/PS1 mice treated or not with PEL24-199. Both in the cortex and hippocampus, the surface of GFAP-positive astrocytes was significantly decreased (n = 4 per group). (**C**) Representative images of GFAP immunostaining in the hippocampus of twelve months-old APP/PS1 treated with water or PEL24-199. Scale bar = 200 µm. (**D**) The surface of GFAP-positive astrocytes was significantly decreased in the hippocampus of PEL24-199 treated animals compared to untreated APP/PS1 animals (n = 5–6 per group). (**E**) Heat-map of activated astrocytes and microglia genes in the cortex and hippocampus of PEL24-199-treated APP/PS1 mice expressed as a percentage of control untreated APP/PS1 (red: increased expression; green: decreased expression). (**F**–**J**) Relative GFAP, C3, Lcn2, Clec7a, and TLR2 gene expression in the hippocampus of APP/PS1 mice treated with PEL24-199 is expressed as a percentage of untreated APP/PS1 (n = 9–13 per group). (**K**) Western blot analysis of GFAP was performed in the cortex and hippocampus of water and PEL24-199 treated APP/PS1 mice. (**L**)**.** Histograms representing GFAP protein levels in the cortex and hippocampus of PEL24-199 treated and untreated APP/PS1 mice (n = 4–6 per group). Significance at the Mann-Whitney statistical test is indicated by *: *p* < 0.05, ** *p* < 0.001, *** *p* < 0.001.

**Figure 4 ijms-24-05285-f004:**
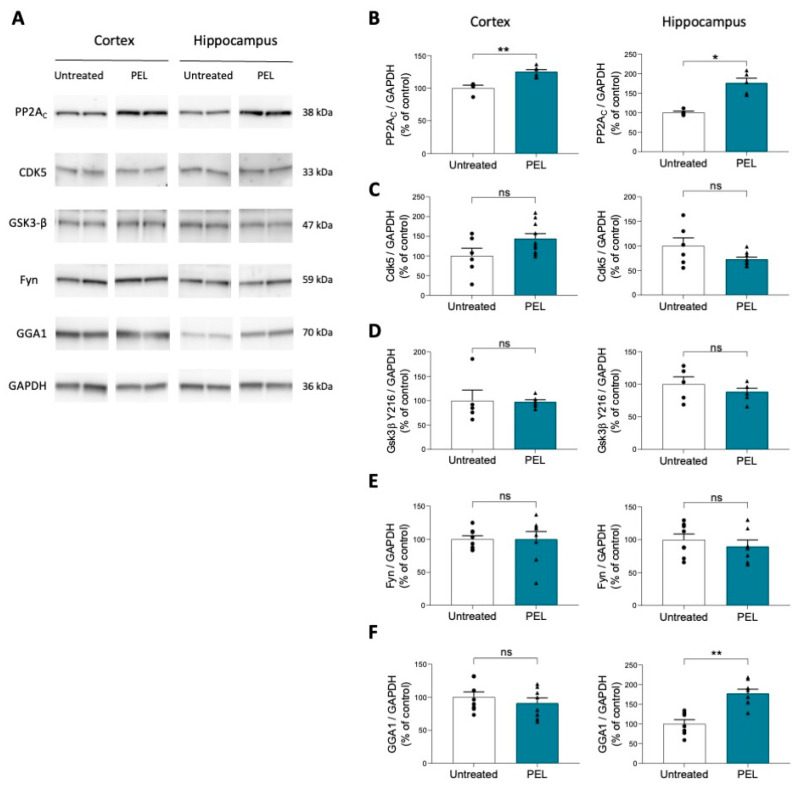
PEL24-199 treatment increases PP2AC and GGA1 protein expression in APP/PS1 mice (**A**). Western blot analysis was performed in the cortex and hippocampus of water- and PEL24-199-treated APP/PS1 mice. (**B**–**F**) Histograms representing PP2AC, Cdk5, Gsk3-β, Fyn, and GGA1 protein levels in the cortex and hippocampus of PEL24-199-treated and untreated APP/PS1 mice. PP2AC expression was significantly increased both in the cortex and in the hippocampus, and GGA1 expression was elevated in the hippocampus. The analysis did not reveal any change in Cdk5, Gsk3-β, and Fyn expression (n = 4–8 per group). Significance at the Mann-Whitney statistical test is indicated by *: *p* < 0.05; **: *p* < 0.01 and *ns* as not significant.

**Figure 5 ijms-24-05285-f005:**
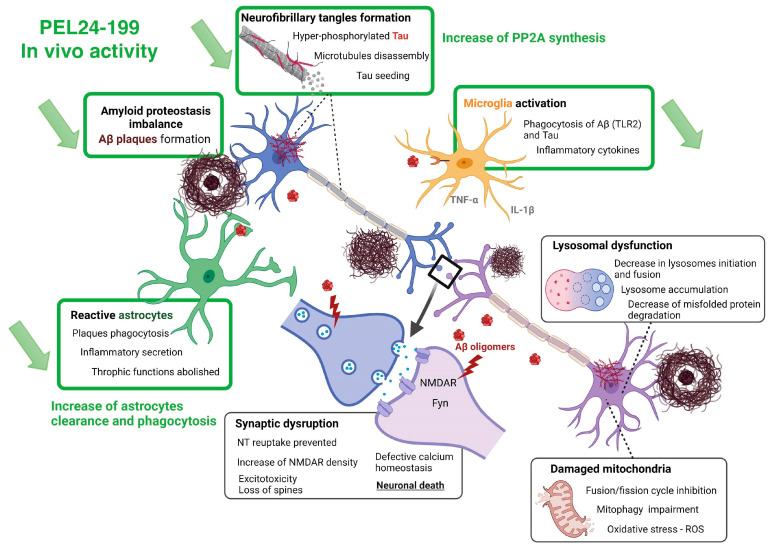
Summary of PEL24-199 in vivo activity and common lesional processes targeted by PEL24-19.

## Data Availability

The data presented in this study are available in Appendix A.

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
