# Peer review of "A Polyaminobiaryl-Based β-secretase Modulator Alleviates Cognitive Impairments, Amyloid Load, Astrogliosis, and Neuroinflammation in APPSwe/PSEN1ΔE9 Mice Model of Amyloid Pathology"

_ijms, 2023, doi:10.3390/ijms24065285_

Round 1
Reviewer 1 Report
In this manuscript, authors have screened a previously validated polyamino biaryl PEL24-199. The manuscript is very well written and presented appropriately. This compound has been mentioned as a β-secretase modulator in cells. They have used appropriate behavioural tests and biochemical assays to validate PEL24-199 as an active compound in restoring cognitive impairments and reducing amyloid load, astrogliosis and neuroinflammation. The manuscript can be accepted after a few revisions. Few queries that need to be addressed are mentioned below.
1. The compound PEL24-199 has been mentioned as a β-secretase modulator in cells. However, the evidence for the same is lacking in the manuscript. The authors need to justify this PEL24-199 property.
2. Although PEL24-199 has an unchanged CTFβ level, it is necessary to show proof of evidence that How PEL24-199 is altering BACE gene or BACE protein levels in the used mice brain.
3. For the reader’s convenience, it is suggested to mention the corresponding behavior test on the top of each graph presented in Fig-1.
Author Response
IJMS Manuscript Ref.: ijms-2159929
First Author: Marie TAUTOU
Reviewer 1 comments:
In this manuscript, authors have screened a previously validated polyamino biaryl PEL24-199. The manuscript is very well written and presented appropriately. This compound has been mentioned as a β-secretase modulator in cells. They have used appropriate behavioral tests and biochemical assays to validate PEL24-199 as an active compound in restoring cognitive impairments and reducing amyloid load, astrogliosis, and neuroinflammation. The manuscript can be accepted after a few revisions. A few queries that need to be addressed are mentioned below.
- The compound PEL24-199 has been mentioned as a β-secretase modulator in cells. However, the evidence for the same is lacking in the manuscript. The authors need to justify this PEL24-199 property.
We assayed BACE1 aspartyl protease enzymatic activity using the Abcam Beta Secretase Activity Assay Kit (Fluorometric, ref. ab65357) adapted to measure BACE1 activity in tissue extracts. We found no difference between PEL24-199 treated and untreated animals. The detailed sequence is not given although the assay is provided to measure BACE1 activity towards BACE1 cleavage of APP between residue 671-672 of APP isoform 695, then corresponding to the beta-site cleavage releasing C99 fragment.
BACE activity was assayed using mouse brain tissue homogenates (n=6 per condition) from the cortex and hippocampus. The histogram represents the ratio between BACE1 activity in untreated animals versus PEL24-199 treated animals. BACE1 activity is not significantly different between treated and untreated animals, either in the cortex or hippocampus.
Therefore, BACE1 aspartyl protease activity is not modified in PEL24-199 treated as previously shown in SY5Y5-cells or direct in vitro investigation. Thus, the modulatory effect is not related to the protease enzymatic activity.
- Although PEL24-199 has an unchanged CTFβ level, it is necessary to show proof of evidence that How PEL24-199 is altering the BACE gene or BACE protein levels in the used mice brain.
BACE expression was assessed by Western blotting.
A slight but not significant decrease of BACE1 expression is noticed in the cortex and hippocampus of PEL24-199 treated animals when compared to the untreated condition.
Neither the BACE1 enzymatic assay nor BACE1 brain tissue expression provides clues to decipher the mechanism of PEL24-199 beta-secretase modulatory activity determined in SY5Y-APP cells. Further investigation would therefore be needed to determine the mode of action of PEL24-199 toward the modulatory activity of BACE1. This modulatory activity could occur toward a preferential substrate recognition, beta-site versus beta’-site, through its cell-trafficking implicating GGA1 (in the present manuscript), post-translational modifications (phosphorylation in relationship with PP2A expression ?), or co-factors, which are currently under investigation.
However, since this is yet rather speculative, an additional sentence has been added at the end of the first discussion paragraph to mention that yet the modulatory activity of BACE1 remains yet undetermined and would necessitate further investigations.
- For the reader’s convenience, it is suggested to mention the corresponding behavior test on the top of each graph presented in Fig-1.
Thanks to the referee for this constructive comment. We have added the behavior test at the top of each graph in figure 1 in the revised manuscript.

Reviewer 2 Report
The MS "A polyaminobiaryl-based β-secretase modulator alleviates cognitive impairments, amyloid load, astrogliosis, and neuroinflammation in APPSwe/PSEN1E9 mice model of amyloid pathology" is really interesting, authors provide a well designed and convincing study.
Here are my comments about the MS.
Intro:
line 45 ref format: (Wang et al., 2019).
Please clearly state the hypothesis in the introduction section
Overall, I find the introduction a bit misleading and hard to follow. There is too many information, without clear links between them, the structure of it doesn't fully present the background and what the study is about.
I would highly suggest to re-write this section.
Materials and methods:
I would move this section after the introduction .
for APPSwe/PSEN1∆E9 mouse could you please add the vendor/cat#
Please could you add a section about the custom macro parameters. It would be interesting to see the parameters used for the analyze.
Results:
fig 1-2 different name for APP
Fig 3: I would suggest to quantify microglia since there is a microgliosis as well. Also, I would suggest to quantify some molecules involved in the phagocytosis pathways. Lamp2, ...
Fig 4: all along the MS BACE1 is cited but I don't see any direct quantification of the latter. I would suggest to include it .
It would be really interesting to study the blood-brain-barrier. the BBB integrity is critical in AD as we seen lately in clinical trials. Using WB or other methods I highly recommend to evaluate the BBB's junctions such as Claudin 5, ZO-1, Coll-IV, ...
Figure 2, 3 and 4: every group should be on the same gel. The way you present makes think that groups are on different gels. Could you please address this issue.
I don't know if it's because of the formatting from MDPI but all the figures are small especially the representative images.
Discussion:
the -secretase noncompetitive inhibitor com- 323: I think you miss a word here
I would also suggest to add in the discussion section the limitations of this study and also discuss the perspectives of this work
I would suggest to add a figure to summarize the findings.
Thank you
Author Response
IJMS Manuscript Ref.: ijms-2159929
First Author: Marie TAUTOU
Reviewer 2 concerns and answers:
The MS "A polyaminobiaryl-based β-secretase modulator alleviates cognitive impairments, amyloid load, astrogliosis, and neuroinflammation in APPSwe/PSEN1E9 mice model of amyloid pathology" is really interesting, authors provide a well-designed and convincing study.
Here are my comments about the MS.
Intro:
line 45 ref format: (Wang et al., 2019).
Author's response:
Thanks for the detailed reading that the authors acknowledge. The format is corrected in the revised manuscript.
Please clearly state the hypothesis in the introduction section
Author's response:
The hypothesis is now clearly stated in the introduction section.
Based on the structure activity relationship and in vivo beneficial effect of other drugs with chemical skeleton both different from chloroquine and current PEL24-199, we hypothesize that PEL24-199 in vivo activity towards amyloid pathology may rely on chemical properties independent of the lysosomotropic or alkalizing of other former evaluated drugs.
The working hypothesis has been added in the beginning of the last introduction paragraph.
Overall, I find the introduction a bit misleading and hard to follow. There is too many information, without clear links between them, the structure of it doesn't fully present the background and what the study is about.
Authors’ response:
IJMS is not a specialized journal in the field of neurodegenerative disease. Then, in the introduction, the first paragraph give knowledge about Alzheimer’s disease pathophysiology, and giving more attention to the amyloid pathology. As in vitro our drugs are suggested to be modulator of the beta-secretase and that this protease is a key protease of amyloid-beta-peptide, the second paragraph summarize knowledge with regards to what is known about BACE1. The third paragraph provide the knowledge about the know in vitro and in vivo activity of PEL24-199, including its structure activity relationship towards other family of molecules which were derived from the pharmacophore design. Finally, the last paragraph also provide knowledge of the rational for testing our drugs in transgenics models of both Tau and amyloid pathologies, which is definitely not the current strategy used for drug development in the field of Alzheimer’s disease.
We therefore understand that the introduction could appears as dense. We aim to provide as much knowledge as possible considering that IJMS readers are not specialist in the field of AD and even in AD drug development. We nevertheless acknowledge the referee’s for is comment and added a clear hypothesis to the introduction to clarify the objective of the study.
Materials and methods:
I would move this section after the introduction .
for APPSwe/PSEN1∆E9 mouse could you please add the vendor/cat#
Authors’ response:
These were provided a long time ago by Sanofi Company while collaborating with Laurent Pradier (Casas et al., 2004). There is therefore no catalogue origin since we originally contributed to the characterization of the amyloid pathology in this transgenic model compared to the one newly developed and published (Casas et al., 2004). Since then, the transgenic model is bred in the research center, we referred to the reference 36
Please could you add a section about the custom macro parameters. It would be interesting to see the parameters used for the analyze.
Authors’ response:
A paragraph has been added about the custom macro in the revised manuscript.
A custom macro derived from the Image J Immunohistochemistry Image Analysis Toolbox was used (https://imagej.nih.gov/ij/plugins/ihc-toolbox/index.html). After adapting the DAPI contrast, the region of interest was manually defined. Blind to the images’ identification, tests were run on a few representative Ab-labeled and Thioflavine-stained images to define the threshold to apply that encompasses the greatest number of plaques without including non-specific signal. The same procedure was run for the GFAP astrocytes staining. These defined thresholds were applied to all images and executed automatically. The 12 measured parameters were as follow : Total surface of the area in mm², Total number of plaques, Plaques total surface in mm², Number of plaques 2-150 µm², Number of plaques 150-500 µm², Number of plaques 500-1500 µm², Ratio plaques area per mm², Ratio number of plaques per mm², GFAP total surface in mm², Ratio GFAP area per mm².
Results:
Fig 1-2 different name for APP
Authors’ response:
Instead Tg in figure, we have replaced by APP/PS1.
Fig 3: I would suggest quantifying microglia since there is a microgliosis as well. Also, I would suggest quantifying some molecules involved in the phagocytosis pathways. Lamp2, ...
Authors’ response:
Lamp2, p62, LC3, TFEB were performed but did not provided evidence of clear expression changes.
From our experience, expression difference in whole brain tissue lysate should be over 10 to 15% to readily objective a difference. Therefore, more quantitative methods should be used or even immunocytofluorescence using tissue clearing should be alternative approaches to be considered instead of the gross western-blot semi-quantitative method. We therefore consider that these results do not provide clear conclusion about any functional changes of the endolysosome and autophagy pathway using solely western-blots and data were not added to the manuscript. Raw data are at the disposal of the referee if requested.
Fig 4: all along the MS BACE1 is cited but I don't see any direct quantification of the latter. I would suggest including it.
Authors’ response:
This concern is also mentioned by the first referee and please find our answer.
We assayed BACE1 aspartyl protease enzymatic activity using the Abcam Beta Secretase Activity Assay Kit (Fluorometric, ref. ab65357) adapted to measure BACE1 activity in tissue extracts. We found no difference between PEL24-199 treated and untreated animals. The detailed sequence is not given although the assay is provided to measure BACE1 activity towards BACE1 cleavage of APP between residue 671-672 of APP isoform 695, then corresponding to the beta-site cleavage releasing C99 fragment.
BACE activity was assay using mouse brain tissue homogenates (n=6 per condition) from the cortex and hippocampus. The histogram represents the ratio between BACE1 activity in untreated animals versus PEL24-199 treated animals. BACE1 activity is not significantly different between treated and untreated animals, either in the cortex or hippocampus.
Therefore, BACE1 aspartyl protease activity is not modified in PEL24-199 treated as previously shown in SY5Y5-cells or direct in vitro investigation. Therefore, the modulatory effect is not related to the protease enzymatic activity.
BACE expression was assessed by Western blotting.
A slight but not significant decrease of BACE1 expression is noticed in the cortex and hippocampus of PEL24-199 treated animals when compared to the untreated condition.
Neither the BACE1 enzymatic assay nor BACE1 brain tissue expression provide clues to decipher the mechanism of PEL24-199 beta-secretase modulatory activity determined in SY5Y-APP cells. Further investigation would therefore be needed to determine the mode of action of PEL24-199 towards the modulatory activity of BACE1. This modulatory activity could occur toward a preferential substrate recognition, b-site versus b’-site, through its cell-trafficking implicating GGA1 (in the present manuscript), post-translational modifications (phosphorylation in relationship with PP2A expression ?), or co-factors, which are currently under investigation.
However, since this is yet rather speculative, an additional sentence has been added at the end of the first discussion paragraph to mention that yet the modulatory activity of BACE1 remains yet undetermined and would necessitate further investigations.
It would be really interesting to study the blood-brain-barrier. the BBB integrity is critical in AD as we seen lately in clinical trials. Using WB or other methods I highly recommend evaluating the BBB's junctions such as Claudin 5, ZO-1, Coll-IV, ...
We entirely agree with the reviewer. The BBB integrity should be investigated. This will be included in future studies since, in the present study this is out of the scope of the present manuscript which aim to provide insights of the potential interest of PEL24-199 as a drug efficient for both Tau and amyloid pathology. We completely agree that the mode of action, as well as other potential effect must be uncovered. This is currently undertaken through the identification of the biological target of our drug. For these reasons, we won’t consider the BBB functionality that can also be investigated using hemosiderin used in vivo to assess the gross impermeability maintenance of the BBC.
Figure 2, 3 and 4: every group should be on the same gel. The way you present makes think that groups are on different gels. Could you please address this issue.
Authors’ response:
As for the figure provide in response to the referee for BACE1. Groups are loaded on the same gel and gel are repeated minimally twice. As shown on the graph a minimum of six samples are loaded per condition and we decided to present only two lane per group that were from the same gel. The raw data are at the disposal of the reviewers and are regroup on a single file of 162 slides.
I don't know if it's because of the formatting from MDPI but all the figures are small especially the representative images.
Authors’ response:
Yes, the format is given by IJMS authors’ instructions.
Discussion:
the -secretase noncompetitive inhibitor com- 323: I think you miss a word here
Authors’ response:
We acknowledge the reviewer for is detailed reading of our manuscript. This mistake is now corrected in the revised version.
I would also suggest adding in the discussion section the limitations of this study and discuss the perspectives of this work.
Authors’ response:
The limitations of the present study are added in the conclusions section especially referring to the lack of a clear relationship with BACE1 which would definitely need further investigation.
I would suggest adding a figure to summarize the findings.
Authors’ response:
We have added a synthetic figure, thanks for suggesting this improvement.
Thanks to the referee to is very constructive comment and we hope that we have provide answers and additional data to support the conclusion of the present study.

Round 2
Reviewer 2 Report
thank, I endorse the MS